# The Content of Phenolic Compounds and Organic Acids in Two *Tagetes patula* Cultivars Flowers and Its Dependence on Light Colour and Substrate

**DOI:** 10.3390/molecules27020527

**Published:** 2022-01-14

**Authors:** Agnieszka Krzymińska, Barbara Frąszczak, Monika Gąsecka, Zuzanna Magdziak, Tomasz Kleiber

**Affiliations:** 1Department of Ornamental Plants, Dendrology and Pomology, Poznań University of Life Sciences, 60-594 Poznań, Poland; 2Department of Vegetable Crops, Poznań University of Life Sciences, 60-594 Poznań, Poland; barbara.fraszczak@up.poznan.pl; 3Department of Chemistry, Poznań University of Life Sciences, 60-625 Poznań, Poland; monika.gasecka@up.poznan.pl (M.G.); zuzanna.magdziak@up.poznan.pl (Z.M.); 4Laboratory of Plant Nutrition, Department of Plant Physiology, Poznań University of Life Sciences, 60-198 Poznań, Poland; tomasz.kleiber@up.poznan.pl

**Keywords:** chemical composition, edible flowers, *Tagetes*, LEDs

## Abstract

The main focus of the study was to determine the content of phenolic acids, flavonoids, and organic acids in the flowers of *Tagetes patula* ‘Petite Gold’ and ‘Petite Orange’. The growth of the plants was assessed depending on the cultivation conditions. The above plants were illuminated with white light, whereas the ‘Petite Gold’ ones with white light enhanced with blue or red light. Both cultivars grew in a two-level-mineral compounds organic substrate. The research showed that the French marigold flowers were rich in phenolic compounds and organic acids. The ‘Petite Gold’ flowers had more bioactive compounds compared with the ‘Petite Orange’ flowers. Three flavonoids, 10 phenolic acids and seven organic acids were found in the ‘Petite Gold’ flowers. The artificial lighting used during the cultivation of the plants showed diversified influence on the content of organic compounds in their flowers. The measurements of the plants’ morphological traits and the number of inflorescences showed that illumination with red light resulted in a better effect. Large plants with numerous inflorescences grew in the substrate with a lower content of nutrients.

## 1. Introduction

It is commonly know that flowers enhance the aesthetics of the environment. Some of them are edible and can be used in two ways. They increase both the decorative and nutritional value of food since they contain chemical components valuable for human health [1,2] e.g., phenolic acids, flavonoids, and tannins [2,3].

One of those valued edible flower [4] plants is the French marigold (*Tagetes patula* L.). Its flowers contain bioactive compounds from different groups, including carotenoids, flavonoids, triterpene, alkaloids, and saponins [5]. The plant’s flowers exhibit antibacterial, antifungal, anti-inflammatory, antioxidative as well as hypertensive and hypotensive activities [6]. However, according to Egebjerg et al. [7], both phytochemical and toxicological data are insufficient to determine the amount of flowers that can be safely consumed by humans.

Flowers for consumption are traditionally cultivated and harvested from fields. However, the process is both weather and season-conditioned. French marigold belongs to those plants that can be successfully grown in greenhouses ensuring the year-round harvesting. However, in advanced production technologies artificial lighting, usually led-type, is used to ensure constant conditions of cultivation, regardless of the season. A correct selection of artificial lighting colour and intensity results in a good-quality high yield. Light is a very important environmental factor influencing the growth and production of secondary metabolites in plants [8,9,10]. In general, the light wavelength, its intensity, and the time of illumination significantly affect the accumulation of metabolites in plants [11]. Red (R) light stimulates photosynthesis and plants growth [12]. Blue (B) light influences the concentration of chlorophyll, photomorphogenesis and the accumulation of antioxidants [13]. The amount of primary and secondary metabolites such as sugars, vitamin C, flavonoids, and polyphenols increases along with the amount of R and B lights in the spectrum [14]. For example, a study on *Anoectochilus roxburghii* showed that the exposure of the plants to the RB light spectrum (80%:20%) resulted in a higher flavonoid content than the exposure to white light (W), R light, B light, and WRB light [15].

The selection of the right substrate is of key importance for the cultivation of French marigold [16]. The substrate should ensure optimal air-water relations as well as chemical properties (pH, salinity, content of nutrients). High moor peat substrates are used by standard for marigold cultivation [17]. However, there are no scientific publications providing data on the chemical composition of the substrate for the production of French marigold for edible flowers grown under artificial lighting.

The aim of the study was to determine the influence of the colour of light and the chemical composition of the peat substrate on the yield of *Tagetes patula* ‘Petite Gold’ flowers as well as their content of phenols and organic acids. The content of phenols in the *Tagetes patula* ‘Petite Gold’ and ‘Petite Orange’ flowers grown under W light was also compared.

## 2. Results and Discussion

The first measurement taken after 30 days of growth did not reveal differences in the height and diameter of the plants, regardless of the cultivar and substrate (Figure 1A,B). The second measurement taken after 90 days showed that the plants of the ‘Petite Orange’ cultivar growing in substrate I were taller and had greater diameters than those growing in substrate II. The substrate used for the cultivation significantly diversified the number of inflorescences (Figure 1C). The plants of both cultivars growing in substrate I had more inflorescences at both terms, which may have indicated an excessively rich chemical composition of substrate II. However, Król [18] observed that nitrogen fertilisation positively influenced the number of flower heads per plant. There were more ‘Petite Orange’ than ‘Petite Gold’ inflorescences.

Light significantly diversified the growth of the ‘Petite Gold’ plants (Figure 2A,B). In both substrate combinations and at both measurement dates the tallest plants grew under W + R light, whereas the W + B light application resulted in the shortest plants. Similar observations were defined concerning the plant diameter. Differences in the dynamics of plant growth resulted in differences in the N-NO_3_ uptake, which was five times smaller in the plants grown under W + B light than in those grown under W + R light. These results confirm the thesis that B light inhibits the elongation growth of various plant species [19,20,21,22]. B light inhibits cell growth, whereas B light photoreceptors can regulate and change the expression through which the stem elongation growth is inhibited [23]. However, studies on sunflower [24] and marigold [25] showed that R light had greater influence on plant compactness than B light. This effect may have been caused by the absence or a small amount of B light in the spectrum. Interestingly, in the conducted study, after 30 days of the experiment, the plants growing under W + R light on the substrate with a higher NPK content were smaller with smaller diameters compared to the plants growing on substrate I. According to Bergstrand et al. [26], this may have been caused by a higher accumulation of biomass in the plants growing under W + R light.

Substrate II also influenced the number of inflorescences in the plants growing under W light (Figure 2C). Both measurements showed that the plants growing under W light and in substrate II had much fewer inflorescences than the plants growing under W light but in substrate I. It is noteworthy that the plants growing in substrate II were characterised by a much lower P and K uptake (statistically insignificant differences) than the plants growing in substrate I. Both elements are very important in the development of the generative phase of plants. W + B light also negatively affected the number of inflorescences during the entire experiment. The plants growing under W + R light had over four times more inflorescences than those growing under W + B light. Red light stimulates the flowering of various plant species [27,28,29]. In the undertaken experiment B light noticeably inhibited the development of inflorescences probably because this light generally inhibits the growth and development of plants. Park and Runkle [30] indicate that a small amount of B light (5–25%) in the entire spectrum may stimulate the growth and flowering of plants. This fact was confirmed by our study, where the number of inflorescences in the plants growing under W light was similar to the number of inflorescences in the plants growing under R light although W light contained more than two times less R light; yet, the sum of B light was similar for both spectra. The higher dose of R light in the spectrum significantly affected the production of inflorescences at high nitrogen fertilisation. Nitrogen stimulates, to a high degree, the excessive vegetative growth of plants. Despite this, R light influenced the formation of inflorescence buds.

The study showed that the factors under analysis significantly influenced changes in the chemical composition of the substrate (Table 1 and Table 2). The substrate samples collected from under the plants grown under different lighting conditions significantly differed from each other in the content of most macronutrients (N-NO_3_, P, K, Ca, Mg, S-SO_4_) and micronutrients (Mn, Zn, and Cl). Different levels of nutrition caused significant differences in the content of P, K, and Mg. Generally there were no significant changes in the content of micronutrients. Meanwhile, in case of EC, an increasing tendency (but not proved significantly) was found for substrate with higher content of nutrient—the relations were determined also in case of each light colour. While analysing light colour effect, an increasing tendency was found of EC for W + B—but the lowest EC values were observed for W and W + R colour. The changes in the content of nutrients resulted from their uptake by plants. This means that the factors under analysis may significantly influence both the growth and yield quality. Kopsell and Sams [8] observed that additional illumination with B light for five days before harvest significantly increased the content of macronutrients (Ca, P, K, Mg, and S) and micronutrients (Cu, Fe, Br, Mn, Mo, and Zn) in the shoot tissues of sprouting broccoli microgreens.

Edible flowers contain various bioactive compounds, including phenolic compounds [3]. In the conducted study the total phenolic content (TPC) ranged from 1.12 to 1.55 mg g^−1^ FW (Table 3). There were no cultivar- or substrate-dependent differences observed. Only the ‘Petite Gold’ plants growing in substrate II and exposed to B + W light differed significantly in the TPC (the highest value—1.35 mg g^−1^) from the plants exposed to the R + W light (the lowest value—1.12 mg g^−1^).

The flavonoid profile of the *Tagetes patula* flowers of the ‘Petite Orange’ and ‘Petite Gold’ cultivars grown on different substrates under W light included catechin, quercetin, and rutin (Table 3). Similar results were obtained in the study conducted by Youssef et al. [31]. Quantitative changes were observed in the flavonoid profile. While quercetin was the dominant compound in all the combinations. The catechin content in the ‘Petite Gold’ plants growing in substrate II under W light increased significantly up to 6.43 µg· g^−1^ FW. The quercetin content in the plants of both cultivars growing in substrate I was significantly higher (869.3 µg· g^−1^ FW in ‘Petite Orange’ and 788.9 µg· g^−1^ FW in ‘Petite Gold’) than in the plants cultivated in substrate II. The rutin content was very diverse—it was higher in the ‘Petite Gold’ plants. The highest rutin content was found in the flowers of the plants cultivated in substrate II (32.78 µg· g^−1^ FW). The total flavonoid content in the plants of both cultivars growing in substrate I was significantly higher than in the plants growing in substrate II. The mean values of the content of each flavonoid showed that there were no significant differences between the cultivars. The mean values of the catechin and rutin content in the plants cultivated in substrate II were significantly higher, whereas the mean content of quercetin and total flavonoids was higher in the plants cultivated in substrate I.

The flavonoid profile of the *Tagetes patula* ‘Petite Gold’ plants showed that different lighting conditions affected the content and synthesis of flavonoids (Table 3). The highest catechin content was found in the plants growing under W + B light in substrate II, whereas the lowest content was found in the plants growing under W and W + R lights in substrate I. Earlier studies also showed that the addition of B light also increased the content of phenolic acids and flavonoids. For example, the addition of B light to in-vitro cultures of chokeberry shoots caused the content of individual phenolic acids and their total content to increase several times [32]. The exposure of the flower stalk of Chinese kale to low intensity B light (50 μmol m^−2^ s^−1^) during storage also resulted in a significant increase in the content of phenolic acids and flavonoids, as compared with dark storage [33].

Fu et al. [34] conducted a study on tobacco and observed that the content of methyl flavonoid derivatives was positively correlated with the proportions of far red light (FR; 716–810 nm) and near infrared (NIR; 810–2200 nm) in the sunlight spectrum but negatively correlated with the proportion of ultraviolet radiation (UV-A; 350–400 nm) and the red to far red ratio (R/FR). The content of flavonoid glycoside derivatives was positively correlated with the UV-A proportion and R/FR ratio, and negatively correlated with FR and NIR. These authors’ observations were confirmed in our research, where the plants cultivated under W + R light in both substrates had a significantly higher content (even three times greater) of quercetin (methyl derivative). Rutin (glycoside derivative) was not detected under W + B light in substrate II, whereas the highest content was found in the plants growing under W light in substrate II. It is likely that W light with a smaller amount of R light had the least inhibitory effect on rutin synthesis. Light also affected the total content of flavonoids, which was significantly greater (even three times) under W + R light. It was related to the high share of methyl flavonoid derivatives in profiling. The quality of light with higher shares of FR and NIR increases the activity of flavonoid methyltransferases, but inhibits the activity of flavonoid glycosyltransferases. A high share of UV-A and a high R/FR ratio may increase the activity of flavonoid glycosyltransferase, but it may also inhibit the activity of flavonoid methyltransferase [34,35]. It is also important to note that the influence of light quality on the production of flavonoids in plants is also species—and even cultivar—dependent [36].

The profile of phenolic acids in both cultivars was highly diversified. It included ten acids (Table 4). The ‘Petite Gold’ plants exposed to W light had a richer phenolic acid profile than the ‘Petite Orange’ plants. 4-hydroxybenzoic acid (4-HBA), gallic, and vanillic acids were identified in both cultivars and substrates. 2.5-dihydroxybenzoic acid (2.5-DHBA), protocatechuic and syringic acids were detected only in the ‘Petite Gold’ plants. The sum of phenolic acids was higher in the ‘Petite Gold’ plants than in the ‘Petite Orange’ plants. Additionally, in substrate II the sum of phenolic acids was higher than in substrate I. The mean content of nearly all acids (excluding chlorogenic and vanillic acids) was significantly higher in the ‘Petite Gold’ plants than in the ‘Petite Orange’ plants. The light colour noticeably influenced the phenolic acid profile in the ‘Petite Gold’ plants. The plants exposed to W light and growing in substrate I had no caffeic or chlorogenic acids, whereas those growing on substrate II did not contain trans-cinnamic or sinapic acids. The plants exposed to W + B light and growing in substrate I had no chlorogenic or protocatechuic acids, whereas the plants exposed to W + R light and growing in substrate II did not contain caffeic, chlorogenic or sinapic acids. 2.5-DHBA was the dominant acid in almost all combinations (except for W light and substrate I and W + R light and substrate I). There were significant differences in the content of most acids between the plants exposed to different light colours and cultivated in different substrates. However, there were no significant changes in the content of sinapic and vanillic acids.

The mean acid levels in each light combination showed differences in the content of 2.5-DHA, chlorogenic, trans-cinnamic, gallic, protocatechuic, sinapic, and syringic acids. The mean levels of each compound in different substrates showed the high content of trans-cinnamic, gallic, sinapic, and syringic acids in substrate I and 2.5-DHBA and total phenolic acids in substrate II.

The analysis of the phenolic composition revealed the presence of flavonoids, phenolic acids, both the derivatives of hydroxybenzoic and hydroxycinnamic acids. The content ranged from 1.24 µg· g^−1^ FW for 4-HBA to 1.574 µg.g^−1^ FW for quercetin. The composition of phenolic acids was very similar to the results of the study by Ayub et al. [37].

Significant differences were observed in the profile and content of organic acids in the flowers of *Tagetes patula* ‘Petite Gold’ exposed to different light spectra and grown on different substrates (Table 5). Seven low-molecular weight organic acids were identified. And their content depended on the cultivation conditions. The following acids were found in the flowers of the plants exposed to W light and grown in substrate I: citric, malic, malonic, and quinic. The plants grown in substrate II also contained acetic, fumaric, and succinic acids. The flowers cut from the plants cultivated in substrate II had a much higher total content of these acids, where citric, fumaric and malonic acids were the dominant types.

The total content of all acids identified in the *Tagetes patula* ‘Petite Gold’ plants exposed to W + B light was significantly higher than in the plants exposed to W light, and the plants growing in substrate II had a higher content of these acids compare to the plants growing in substrate I. Acetic, malic, and malonic acids were the dominant types. The total content of all acids identified in the flowers of the plants exposed to W + R light and cultivated in both substrates was similar.

The flowers of the ‘Petite Gold’ cultivar had more organic acids than those of ‘Petite Orange’ cultivar. The differences in the content of organic acids, which were also observed in the flowers of other plants [38,39], indicate that the results may depend on the species and cultivar as well as the type of substrate and lighting, as was the case in our experiment. The analysis showed that the *Tagetes patula* flowers were a good and balanced source of organic acids and phenolic compounds.

## 3. Materials and Methods

### 3.1. Experimental Materials and Design

The experiment was conducted twice—in 2018 and 2019—on two French marigold cultivars: ‘Petite Gold’ and ‘Petit Orange’, with yellow and orange flowers, respectively.

The plants were cultivated in growth chambers under LEDs. Before transplantation into a permanent place the plants had been grown under W light only. Seeds were sown into boxes, and after about four weeks the seedlings were pricked out. After another four weeks the plants were planted in pots and placed in chambers with a diverse light spectrum, where they grew for three months.

Afterwards they were exposed to different lighting—W light as well as W light enhanced with B or R light. The photosynthetic photon flux density (PPFD) from the top of the plants amounted to about 170 μmol m^−2^ s^−1^ (± 14 _SD_) for W light and additionally 60 μmol m^−2^ s^−1^ (± 8 _SD_) for B and R lights. The PPFD for W + R and W + B light combinations amounted to about 230 μmol m^−2^ s^−1^. The daily light integral amounted to about 9.8 mol m^−2^ d^−1^ for W light and to about 13.2 mol m^−2^ d^−1^ for W + R and W + B lights. The PPFD was measured with a PAR-10 quantum sensor (Sonopan, Białystok, Poland). The spectral distribution of light treatments was measured with a BLACK-Comet CXR UV-VIS spectroradiometer (280–900 nm, StellarNet Inc., Tampa, FL, USA). The measurements were taken 15 cm under the lamps approximately at the height of the plant tops. As the plants grew, the lamps were gradually raised to a higher position. Table 6 shows the spectral characteristics of the lamps. The temperature at the preliminary stage was 20 °C and during the experiment it was 18 °C. The plants were watered when necessary.

The vegetation experiments were conducted with peat moss, limed to a pH of 6.50 (in H_2_O) on the basis of the neutralisation curve. The effects of increasing macroelement levels (denoted as I, II) while maintaining a constant quantitative N:P:K ratio of 1.0:0.75:1.25 were investigated. The content of nutrients was as follows: I: N 150, P 112, K 187 mg∙dm^−3^; II: N 200, P 150, K 250 mg∙dm^−3^. A standard content of microelements in the substrate was used: Fe 50.0, Mn 10.0, Zn 10.0, Cu 2.0, B 0.5 mg∙dm^−3^. The plants were not fed during the experiment.

Substrate samples were collected after completion of the vegetation experiments and analysed chemically with the universal method [40]. Macronutrients (N-NH_4_, N-NO_3_, P, K, Ca, Mg, S-SO_4_), Cl and Na were extracted in 0.03 mCH_3_COOH with a quantitative substrate to an extraction solution ratio of 1:10. After the extraction the following measurement methods were applied: N-NH_4_, N-NO_3_—microdistillation according to Bremer with Starck’s modification; P—colorimetry with ammonium vanadomolybdate; K, Ca, Na—photometry; Mg—atomic absorption spectrometry (ASA, on a Carl Zeiss-Jena apparatus, Thornwood, NY, USA); S-SO_4_—nephelometry with BaCl_2_; Cl—nephelometry with AgNO_3_. Micronutrients (Fe, Mn, Zn and Cu) were extracted with Lindsay’s solution containing 5 g EDTA (ethylenediaminetetraacetic acid) in 1 dm^3^; 9 cm^3^ of 25% NH_4_ solution, 4 g citric acid; 2 g Ca(CH_3_COO)_2_·2H_2_O. The content of micronutrients was measured with the ASA method. Salinity was measured conductometrically as an electrolytic conductivity (EC in mS·cm^−1^), whereas pH was measured with the potentiometric method (substrate:water = 1:2) [41].

### 3.2. Morphological Measurements

The first biometric measurements were conducted 30 days after starting the experiment, while the next ones were made another 60 days later. The height and diameter of the plants were measured and the inflorescences were counted.

### 3.3. Sample Preparation

Samples of flowers were collected for analysis about 60 days after the beginning of the experiment under controlled conditions. Phenolic compounds and organic acids were extracted from the homogenised flowers of the two cultivars with 80% ethanol. The samples were sonicated at 40 °C for 20 min and then shaken for 12 h at room temperature. Then they were centrifuged at 3000 rpm, evaporated to dryness and stored at −20 °C before analyses.

### 3.4. Determination of Phenolic Compounds and Organic Acids

Phenolic compounds and organic acids were identified with an ultra-performance liquid chromatography ACQUITY UPLC H-Class System (Waters Corporation, Milford, MA, USA). An ACQUITY UPLC BEH C18 column (2.1 mm 9150 mm, 1.7 lm, Waters) thermostated at 35 °C was used for separation. A mixture of water and acetonitrile (both containing 0.1% formic acid, pH = 2) was used to elution at the flow rate of 0.4 mL· min^−1^ with the gradient elution was used. Peaks were identified by comparing the retention times of chemical standards. A Waters Photodiode Array Detector (Waters Corporation, Milford, MA, USA) was used for the detection of individual compounds at λ = 280 nm and λ = 320 nm.

### 3.5. Determination of Total Phenolic Content (TPC)

The flower extracts were mixed with the Folin-Ciocalteu reagent (diluted with deionised water *v*:*v* (1:1). After 3 min 1 mL of 20% Na_2_CO_3_ was added to the mixtures. Then the samples were kept in darkness for 30 min at room temperature. The absorbance of the samples was measured with a UV spectrophotometer (Carry 300 Bio UV-VisibleSpectrophotometer (Varian, Agilent, Santa Clara, CA, USA) at 765 nm. Gallic acid was used as the standard for TPC quantification. The TPC concentration was expressed as milligrams of gallic acid equivalents per fresh weight (mg GAE· g^−1^FW).

### 3.6. Statistical Analysis

The following factors were analysed in the experiment: the cultivar, the light spectrum (W, W + B, W + R), and the substrate composition or the cultivar and the substrate composition. There were four replicates in each experimental combination and 15 plants in each replicate. The content of phenolic compounds was measured twice and the results were subjected to a three-way analysis of variance. The means were grouped with the Duncan test at a significance level α = 0.05. The results presented in the manuscript are the mean values of the duplicate experiments. The data were analysed statistically with the Statistica program (StatSoft, Kraków, Poland).

## 4. Conclusions

The research showed that the flowers of *Tagetes patula* ‘Petite Gold’ contained more phenolic acids, flavonoids, and organic acids than the ‘Petite Orange’ flowers.

The artificial lighting of the plants during cultivation had a diversified effect on the content of organic compounds in the plants’ flowers. The measurements of the morphological traits of the plants and the number of their inflorescences indicate that the illumination with R light was justified.

The plants cultivated in the substrate with a lower content of nutrients were larger and had numerous inflorescences. However, the plants cultivated in the substrate with a higher content of nutrients had a higher content of phenolic acids in their flowers.

## Figures and Tables

**Figure 1 molecules-27-00527-f001:**
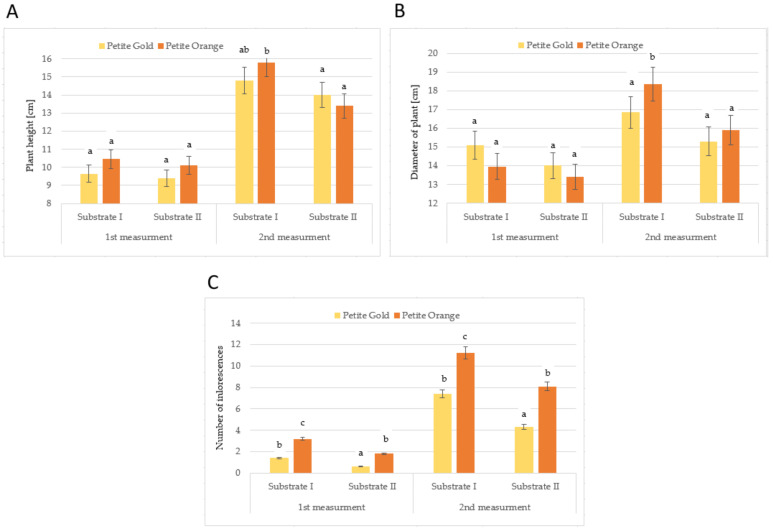
Plant height (**A**), diameter of plants (**B**) and number of inflorescences (**C**) of *Tagetes* ‘Petite Gold’ and ‘Petite Orange’ depending on the substrate used (after exposure to W light). Means followed by the same letters are not significantly different.

**Figure 2 molecules-27-00527-f002:**
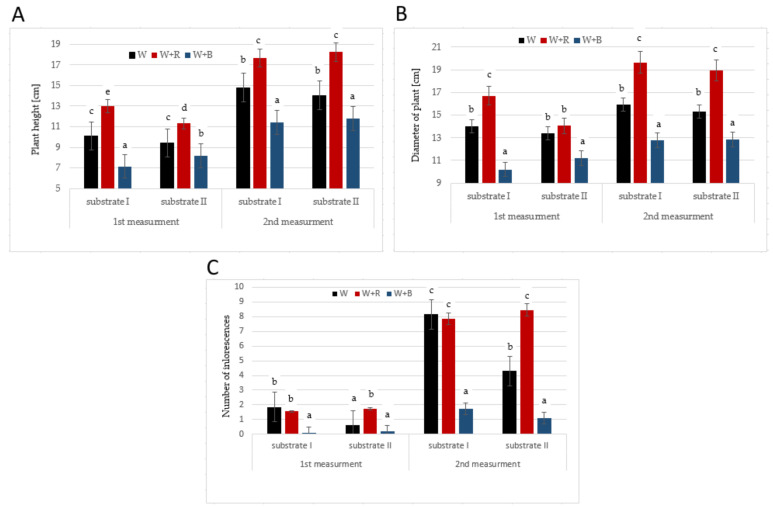
Plant height (**A**), diameter of plants (**B**) and number of inflorescences (**C**) of *Tagetes* ‘Petite Gold’ and ‘Petite Orange’ depending on the substrate and light used. Means followed by the same letters are not significantly different.

**Table 1 molecules-27-00527-t001:** Macronutrients and sodium content in substrate depending on light colour and substrate (mg·dm^−3^).

Light Colour (A)	Substrate (B)	N-NO_3_	N-NH_4_	P	K	Ca	Mg	Na	S-SO_4_
White	I	7.0 a	26.5 a	63.5 a	63.5 a	1246.0 ab	212.0 a	138.0 a	316.5 a
	II	14.0 a	42.0 a	103.5 ab	118.0 ab	1100.0 a	225.5 ab	125.5 a	364.0 ab
White + blue	I	28.0 a	17.5 a	37.5 a	188.0 bc	1401.0 b	244.0 ab	150.0 a	415.0 b
	II	19.5 a	24.5 a	58.5 a	217.0 c	1294.5 ab	262.5 b	136.0 a	388.5 ab
White + red	I	2.0 a	23.0 a	183.0 b	183.0 bc	1246.0 ab	252.5 ab	139.0 a	369.5 ab
	II	7.0 a	26.5 a	63.5 a	63.5 a	1246.0 ab	212.0 a	138.0 a	316.5 a
Mean for A	White	10.5 ab	34.3 a	83.5 a	90.8 a	1173.0 a	218.8 a	131.8 a	340.3 a
	White + blue	23.8 b	21.0 a	48.0 a	202.5 b	1347.8 b	253.3 b	143.0 a	401.8 b
	White + red	4.5 a	24.8 a	123.3 b	123.8 a	1246.0 ab	232.3 ab	138.5 a	343.0 a
Mean for B	I	12.3 a	22.4 a	94.7 b	144.8 b	1297.7 a	236.2 b	142.3 a	367.0 a
	II	13.5 a	31.0 a	75.2 a	132.8 a	1213.5 a	233.3 a	133.2 a	356.3 a

Means in columns followed by the same letters are not significantly different.

**Table 2 molecules-27-00527-t002:** Micronutrients content (mg·dm^−3^) in substrate, pH and EC (mS·cm^−1^) depending on light colour and substrate.

Light Colour (A)	Substrate (B)	Fe	Mn	Zn	Cu	Cl	pH	EC
White	I	13.05 ab	1.45 ab	16.70 a	0.30 a	52.0 a	6.82 ab	0.65 a
	II	12.55 a	1.70 ab	16.60 a	0.30 a	82.0 ab	6.58 a	1.05 ab
White + blue	I	19.10 b	3.20 b	23.65 b	0.40 a	116.5 b	6.61 ab	1.05 ab
	II	13.15 ab	2.40 ab	18.85 a	0.30 a	111.0 b	6.62 ab	1.15 b
Red + white	I	15.10 ab	1.40 b	16.35 a	0.30 a	55.5 a	6.88 b	0.70 ab
	II	11.55 a	1.30 a	15.10 a	0.30 a	92.0 ab	6.73 ab	0.90 ab
Mean for A	White	12.80 a	1.58 ab	16.65 a	0.30 a	67.0 a	6.70 a	0.85 a
	White + blue	16.13 a	2.80 b	21.25 b	0.35 a	113.8 b	6.62 a	1.10 a
	White + red	13.33 a	1.35 a	15.73 a	0.30 a	73.8 a	6.81 a	0.80 a
Mean for B	I	15.75 a	2.02 a	18.90 a	0.33 a	74.7 a	6.77 a	0.80 a
	II	12.42 a	1.80 a	16.85 a	0.30 a	95.0 a	6.64 a	1.03 a

Means in columns followed by the same letters are not significantly different.

**Table 3 molecules-27-00527-t003:** Profiling of flavonoids and total phenolic content of the *Tagetes patula* flowers depending on cultivar and substrate under white light (big letters) and profiling of flavonoids of the *Tagetes patula* ‘Petite Gold’ flowers depending on light colour and substrate (small letters) [µg·g^−1^FW].

Cultivar (A)	Light Colour (B)	Substrate (C, c)	TPC [mg GAE·g^−1^FW]	Catechin	Quercetin	Rutin	Sum
Petite Orange	White	I	1.40 A	1.99 A	869.3 C	3.53 A	874.8 C
		II	1.55 A	2.85 A	180.1 A	3.80 A	186.7 A
Petite Gold	White	I	1.32 A ab	2.33 A a	788.9 C a	11.83 B b	803.1 C a
		II	1.19 A ab	6.43 B b	482.3 B a	32.78 C c	521.5 B a
	White + blue	I	1.19 ab	5.18 b	420.8 a	1.73 a	421.7 a
		II	1.35 b	9.25 c	425.8 a	nd	435.0 a
	White + red	I	1.22 ab	2.87 a	1574.8 c	7.73 b	1585.4 c
		II	1.12 a	5.36 b	1201.5 b	1.25 a	1208.2 b
Mean for A	Petite Orange		1.48 A	2.41 A	524.7 A	3.66 A	530.8 A
	Petite Gold		1.26 A	4.38 B	635.6 B	22.31 B	662.3 A
Mean for B	White		1.25 a	4.38 a	635.6 a	22.31 b	662.3 a
	White + blue		1.27 a	7.22 b	423.3 a	0.87 a	431.4 a
	White + red		1.17 a	4.12 a	1388.2 b	4.49 a	1396.8 b
Mean for C	I		1.36 A	2.16 A	829.1 B	7.68 A	839.0 B
	II		1.37 A	4.64 B	331.2 A	18.29 B	354.1 A
Mean for c	I		1.24 a	3.46 a	928.2 b	7.10 a	938.7 b
	II		1.22 a	7.02 b	703.2 a	11.34 b	721.6 a

Means in columns followed by the same letters are not significantly different.

**Table 4 molecules-27-00527-t004:** Profiling of phenolic acids of the *Tagetes patula* flowers depending on cultivar and substrate under white light (big letters) and profiling of flavonoids of the *Tagetes patula* ‘Petite Gold’ flowers depending on light colour and substrate (small letters) [µg· g^−1^FW].

Cultivar (A)	Light Colour (b)	Substrate (C, c)	2.5-DHBA	4-HBA	Caffeic	Chlorogenic	Trans-Cinnamic	Gallic	Protocatechuic	Sinapic	Syringic	Vanillic	Sum
Petite Orange	White	I II	nd nd	1.32 A 1.62 A	nd 1.17 B	1.78 B 3.05 C	nd nd	2.70 A 4.06 B	nd nd	nd nd	nd nd	1.32 A 5.07 B	7.12 A 14.96 B
Petite Gold	White	I	2.21 A a	1.24 A a	nd	nd	4.38 B b	6.93 C c	1.88 B c	1.71 B b	1.40 B a	4.33 B a	24.10 C a
		II	29.35 B c	6.90 B b	4.54 C c	3.21 C c	nd	3.88 B ab	1.59 B bc	nd	1.29 B a	4.32 B a	55.08 D c
	White + blue	I	28.90 c	4.26 ab	2.57 bc	nd	17.83 d	8.06 c	nd	2.57 b	7.95 c	8.41 a	80.54 d
		II	44.48 d	5.97 ab	1.96 ab	1.53 b	4.04 b	5.84 bc	1.25 b	2.56 b	8.20 c	7.26 a	83.09 d
	White + red	I II	2.71 a 12.82 b	6.86 b 1.47 a	1.43 ab nd	3.98 c nd	3.87 b 11.96 c	7.40 c 2.50 a	3.38 d 2.16 c	1.66 b nd	4.96 b 1.11 a	5.41 a 4.45 a	41.65 b 36.47 b
Mean for A	Petite Orange Petite Gold		nd 15.80 B	1.47 A 4.07 B	0.58 A 2.27 B	2.41 B 1.61 A	nd 2.19 B	3.38 A 5.41 B	nd 1.74 B	nd 0.85 B	nd 1.35 B	3.19 A 4.33 A	11.04 A 39.59 B
Mean for b	White White + blue White + red		15.78 b 36.69 c 7.76 a	4.07 a 5.11 a 4.17 a	2.27 a 2.26 a 0.71 a	1.61 b 0.77 a 1.99 b	2.19 a 10.94 c 7.91 b	5.41 ab 6.95 b 4.95 a	1.74 b 0.62 a 2.77 c	0.85 a 2.56 b 0.83 a	1.34 a 8.07 b 3.04 b	4.32 a 7.84 a 4.93 a	39.59 a 81.82 b 39.06 a
Mean for C	I		1.11 A	1.28 A	nd	0.89 A	2.19 B	4.82 B	0.94 A	0.85 B	0.70 A	2.82 A	15.61 A
	II		14.67 B	4.26 B	2.85 B	3.13 B	nd	3.97 A	0.79 A	nd	0.65 A	4.70 A	35.02 B
Mean for c	I		11.27 a	4.12 a	1.33 a	1.32 a	8.69 b	7.47 b	1.76 a	1.98 b	4.77 b	6.05 a	48.76 a
	II		28.88 b	4.78 a	2.16 a	1.58 a	5.33 a	4.07 a	1.67 a	0.85 a	3.53 a	5.35 a	58.22 b

Means in columns followed by the same letters are not significantly different.

**Table 5 molecules-27-00527-t005:** Profiling of organic acids of the *Tagetes patula* flowers depending on cultivar and substrate under white light (big letters) and profiling of organic acids of the *Tagetes patula* ‘Petite Gold’ flowers depending on light colour and substrate (small letters) [µg· g^−1^FW].

Cultivar (A)	Light Colour (b)	Substrate (C, c)	Acetic	Citric	Fumaric	Malic	Malonic	Succinic	Quinic	Sum
Petite Orange	White	I II	nd nd	nd nd	0.84 A nd	3.63 AB 3.16 AB	4.21 A 4.55 A	nd 3.45 B	0.70 A 2.70 B	9.39 A 14.11 A
Petite Gold	White	I	nd	5.97 B bc	nd	4.85 B a	9.68 B c	nd	1.24 A a	21.74 B a
		II	0.91 Bab	9.26 C cd	9.28 B b	2.65 A a	10.65 B c	0.52 A ab	1.23 A b	34.51 C b
	White + blue	I	2.41 ab	3.91 ab	nd	12.42 b	4.89 ab	2.38 ab	2.63 c	28.64 ab
		II	11.62 c	3.20 ab	nd	16.91 c	8.78 bc	4.74 b	0.98 ab	46.22 c
	White + red	I II	nd 3.15 b	nd 13.93 d	0.37 a nd	6.88 a 3.98 a	11.55 c 2.16 a	19.66 d 9.63 c	0.55 a 2.20 c	39.02 bc 35.06 b
Mean for A	Petite Orange Petite Gold		nd 0.45 B	nd 7.61 B	0.42 A 4.64 B	3.40 A 3.75 A	4.38 A 10.17 B	1.73 B 0.26 A	1.70 A 1.24 A	11.75 A 28.12 B
Mean for b	White		15.78 b 36.69 c 7.76 a	4.07 a 5.11 a 4.17 a	4.64 b nd 0.18 a	1.61 b 0.77 a 1.99 b	2.19 a 10.94 c 7.91 b	5.41 ab 6.95 b 4.95 a	1.74 b 0.62 a 2.77 c	0.85 a 2.56 b 0.83 a
	White + blue									
	White + red									
Mean for C	I		nd	2.99 A	0.42 A	4.24 B	6.95 A	nd	0.97 A	15.56 A
	II		0.45 B	4.63 A	4.64 B	2.91 A	7.60 A	1.99 B	1.97 B	24.31 B
Mean for c	I		11.27 a	4.12 a	0.12 a	1.32 a	8.69 b	7.47 b	1.76 a	1.98 b
	II		28.88 b	4.78 a	3.09 b	1.58 a	5.33 a	4.07 a	1.67 a	0.85 a

Means in columns followed by the same letters are not significantly different.

**Table 6 molecules-27-00527-t006:** Characteristics of white light source.

Light Colour	Wavelength (nm)	PFD * (μmol m^−2^ s^−1^)	% for W Light	% for W + R	% for W + B
UV	320–380	0.5	0.3	0.2	0.2
Violet	380–450	15.4	8.9	6.6	6.6
Blue	450–495	30.3	17.6	13.0	38.9
Green	495–570	53.5	31.1	23.0	23.0
Gold	570–590	18.7	10.9	8.1	8.1
Orange	590–620	21.8	12.7	9.4	9.4
Red (R)	620–700	26.4	15.3	37.2	11.4
Far Red (FR)	700–780	5.6	3.3	2.4	2.4
sum	320–780	172.2	100	100	100
R:FR		4.7	-	15.5	-

* Photon flux density.

## Data Availability

Not applicable.

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
