# Peer review of "The Content of Phenolic Compounds and Organic Acids in Two Tagetes patula Cultivars Flowers and Its Dependence on Light Colour and Substrate"

_molecules, 2022, doi:10.3390/molecules27020527_

Round 1

Reviewer 1 Report

The MS (1521856) titled as "The content of phenolic compounds and organic acids in the flowers of two Tagetes patula cultivars" presented us the influences of both the light scheme and mineral levels on growth as well as the contents of phenolics and organic acids. However, the title did not include the effectors, only mentioning two of the several measured indexex. Secondly, the Tigures and Tables were presented poorly, and the MS was prepared not good. Finally, the data the MS contained were so few. 

Author Response

Thank you for all remarks and comments. Our answers are in the file.

Thank you for all comments and remarks.

Reviewer: The MS (1521856) titled as "The content of phenolic compounds and organic acids in the flowers of two Tagetes patula cultivars" presented us the influences of both the light scheme and mineral levels on growth as well as the contents of phenolics and organic acids. However, the title did not include the effectors, only mentioning two of the several measured indexex.

Answer: The title was completed as below:

The content of phenolic compounds and organic acids in the flowers of two Tagetes patula cultivars depends on light colour and substrate

Reviewer: Secondly, the Figures and Tables were presented poorly, and the MS was prepared not good. Finally, the data the MS contained were so few.

Answer: Figures and tables were prepared according to Instructions for authors.

Reviewer 2 Report

In the attachment file

Author Response

Thank you for all remarks. Our answers are in the file.

Thank you for all comments and remarks.

Reviewer: Abstract should be informative and include the main findings.

Answer: Abstract was enriched.

Reviewer: Language should be revised. It has plenty of grammatical errors as well as sentence-structure mistakes. There are a number of grammatical errors and instances of badly worded/constructed sentences. Besides, some parts are really confused and the text is not always comprehensible. This makes the reader confused, and makes this work uninteresting.

Answer: The manuscript was checked by a qualified translator. If you feel that English is not correct, please give some examples of the wrong wording.

Reviewer: Figures should be represented in higher resolution.

Answer: For reviewed version figures were in manuscript. In finally version figures will be in high resolution.

Reviewer: Introduction should be enriched with recent references.

Answer: Introduction was enriched with references - Meléndez-Martínez et al 2021, Zhang et al 2020, Nissim-Levi et al 2019 and Aliniaeifard et al 2018.

Reviewer 3 Report

A manuscript on the content of phenolic compounds and organic acids in flowers of two Tagetes patula cultivars is introduced. The effect of two different artificial lights and two organic substrates with different mineral content were evaluated in terms of their phenolic acids, flavonoids, and organic acids as well morphological features. The manuscript is well written, the experimental looks sound, affording key results to support valid conclusions. Therefore, the document may be considered for publication in “Molecules” after minor observations made to the authors´ attention as follows:

Line 36 Is it correct that French marigold flowers exhibit hypertensive and hypotensive activities?

Revise it, please

Line 67 As the manuscript does not include a Discussion section, this one should be:

“2. Results and Discussion”  

Lines 229-230 Please integrate this sentence into the previous or the following paragraphs

Line 310, 325,  Along the document use appropriate units, e.g., “min” for minutes, “h” for hours, etc.

Line 332 Use letters for numbers 0-9, e.g., “four replicates”

Author Response

Thank you for all remarks. Our answers are in the file.

Thank you for all comments and remarks.

Reviewer: Line 36 Is it correct that French marigold flowers exhibit hypertensive and hypotensive activities?

Answer: According to literature (Chitrakar et al 2019) it is correct.

Reviewer: Line 67 As the manuscript does not include a Discussion section, this one should be: “2. Results and Discussion”

Answer: The name was changed.

Reviewer: Lines 229-230 Please integrate this sentence into the previous or the following paragraphs

Answer: This sentence was integrated (following paragraphs).

Reviewer: Line 310, 325, Along the document use appropriate units, e.g., “min” for minutes, “h” for hours, etc.

Answer: The changes were done.

Reviewer: Line 332 Use letters for numbers 0-9, e.g., “four replicates”

Answer: The change was done.

Reviewer 4 Report

Article

The content of phenolic compounds and organic acids in the flowers of two Tagetes patula cultivars

Agnieszka Krzymińska, Barbara Frąszczak, Monika Gąsecka, Zuzanna Magdziak and Tomasz Kleiber

Comments and Suggestions for Authors

The main goal of the study was investigating the influence of the light’s color on the growth of two Tagetes patulacultivars and different secondary metabolite compositions. The two variants of the substrate’s chemical composition were used under experimental conditions and the yield of plants was compared.

A correlation analysis was performed between the content of phenolic compounds and organic acids and the colors of light. The study determined the variations of metabolites content in plants cultivated in the substrate with lower or higher contents of nutrients.

The experimental part of the study was well designed and had proper analytical tools and data analysis.

There are several points that need to be addressed

124 please insert EC details.

190-191 I think some corrections are necessary for a clearer understanding. It is stated that” The total content of phenolic acids was higher in the ‘Petite Gold’ plants and substrate II than in the ‘Petite Orange’ plants and substrate I”, but at the 132-133 paragraphs it is indicated at the total phenolic content” There were no cultivar - or substrate-dependent differences observed”.

192-194 The paragraph was not understood: „was significantly higher in the ‘Petite Orange’ plants than in the ‘Petite Orange’ plants”.  What was compared?

Tables 1 and 2 include information regarding the macronutrients and micronutrients content in substrates. More information about the analysis would be preferable. Is it recorded after 30 or 90 days, under experimental conditions, of the “Petite Gold” or “Petit Orange exposures?

I think the experimental design is clear, but the results and the table format are not appropriate for the following they goal. The tables including much data, they are difficult to follow. I suggest using another design, for example graphics.

I suggest revising the sentences for a decrease in the similarities recorded (20%).

Author Response

Thank you for all remarks. Our answers are in the file.

Round 2

Reviewer 1 Report

The reply by the authors did not meet the reviewer's comments.

Reviewer 2 Report

The manuscript is good know. I suggest for publication in the journal.